# Integrated Local and Systemic Communication Factors Regulate Nascent Hematopoietic Progenitor Escape During Developmental Hematopoiesis

**DOI:** 10.3390/ijms26010301

**Published:** 2024-12-31

**Authors:** Carson Shalaby, James Garifallou, Christopher S. Thom

**Affiliations:** 1Division of Neonatology, Children’s Hospital of Philadelphia, Philadelphia, PA 19104, USA; 2Department of Pediatrics, Perelman School of Medicine, University of Pennsylvania, Philadelphia, PA 19104, USA

**Keywords:** hematopoiesis, blood, stroma, endothelium, signaling

## Abstract

Mammalian blood cells originate from specialized ‘hemogenic’ endothelial (HE) cells in major arteries. During the endothelial-to-hematopoietic transition (EHT), nascent hematopoietic stem cells (HSCs) bud from the arterial endothelial wall and enter circulation, destined to colonize the fetal liver before ultimately migrating to the bone marrow. Mechanisms and processes that facilitate EHT and the release of nascent HSCs are incompletely understood, but may involve signaling from neighboring vascular endothelial cells, stromal support cells, circulating pre-formed hematopoietic cells, and/or systemic factors secreted by distal organs. We used single cell RNA sequencing analysis from human embryonic cells to identify relevant signaling pathways that support nascent HSC release. In addition to intercellular and secreted signaling modalities that have been previously functionally validated to support EHT and/or developmental hematopoiesis in model systems, we identify several novel modalities with plausible mechanisms to support EHT and HSC release. Our findings paint a portrait of the complex inter-regulated signals from the local niche, circulating hematopoietic/inflammatory cells, and distal fetal liver that support hematopoiesis.

## 1. Introduction

The hematopoietic stem cells (HSCs) that support lifelong blood cell production are first produced in major arteries at the onset of cardiac function during embryonic development [1]. Local and systemic signaling mechanisms that support developmental hematopoiesis are incompletely understood. Recent studies have begun to elucidate interactions from non-hematopoietic stromal cells and circulating factors that regulate HSC production [2,3]. Single cell transcriptomics have recently profiled embryonic cells during HSC generation, with a focus on developmental hematopoietic processes [4]. This study was designed to leverage those single cell data to determine signaling mechanisms that impact hematopoietic cell development, focusing on cellular factors both within the hematopoietic niche and across tissues that facilitate HSC formation during embryogenesis.

Hematopoiesis occurs in waves during embryonic development, which have been studied using multiple model systems [1,2,3,4,5,6,7]. In all cases, hematopoietic cells are generated from specialized precursors called ‘hemogenic’ endothelial cells (HECs). The first and second waves produce primitive red blood cells, megakaryocytes, and macrophages from yolk sac endothelium. A third wave, which occurs in the dorsal aorta-gonad-mesonephros (AGM) region and other major arteries (~4.5 weeks post conception in humans), produces short term multipotent progenitor cells and long-term hematopoietic stem cells [4,7]. These ‘definitive’ cells transiently colonize the human fetal liver from weeks 4.5–6, and ultimately engraft in bone marrow from weeks 8–12 to support lifelong hematopoiesis [8]. Definitive HECs morph into HSCs through an endothelial-to-hematopoietic transition (EHT) and bud off from the aortic endothelial lining. Cells and mechanisms that facilitate HSC ‘escape’ from the endothelial lining are incompletely understood.

Hematopoietic cell-autonomous transcriptional regulatory circuits control HEC specification and fate, including Notch, Wnt, and TGFβ [5,6]. In addition, stromal and mesenchymal cells in the arterial niche can support HEC and HSC formation. For example, murine Ng2^+^Runx1^+^ perivascular smooth muscle cells in close contact to HECs can support hematopoietic development [3,6]. Indeed, explanted AGM tissue can support EHT and HSC formation [9,10]. Factors in the embryonic circulation also provide signaling input to facilitate HEC specification and HSC formation. Exposure to inflammatory signals, including macrophage- or neutrophil-derived Tumor Necrosis Factor α (TNFα) and via activation of the NFκB, facilitates HEC specification in zebrafish [11,12]. Augmented inflammatory signals can enhance hematopoiesis from cultured stem cells [13]. Less well studied are how signaling post-HEC specification facilitates the escape of nascent HSCs from the arterial endothelial lining.

We hypothesized that resolving local and systemic signals that impact human developmental hematopoiesis at single cell resolution would facilitate novel insight into factors that converge to regulate definitive HEC and HSC formation in the AGM, including the initiation and propagation of mechanisms necessary for nascent HSC release. Indeed, recent studies leveraged single cell transcriptomic data from primary human embryonic cells to identify novel cell-intrinsic pathways that regulate HSC formation [14,15] or facilitate development of in vitro-derived HSCs [16,17]. We identified established signaling networks and elucidated novel factors from local and systemic sources, including the fetal liver, that direct human developmental hematopoiesis. We additionally identify plausible mechanisms by which inflammatory signaling contributes to EHT, including transcriptional output from signaling events that facilitates cell migration and escape. Harnessing these signaling modalities and mechanisms may boost the quantity or quality of HSCs by better recapitulating the hematopoietic niche in vitro.

## 2. Results

### 2.1. Single Cell Analyses Identify Cells and Interactions That Support Hematopoiesis in the AGM Microenvironment

We reprocessed single cell RNA sequencing (scRNAseq) data from human embryonic AGM cells collected at 4.5–6 weeks post-conception [4] using Seurat (v5 [18]). In addition to HECs, HSCs, and mature hematopoietic cells, we identified and annotated stromal and endothelial cell populations [19] (Figure 1A,B). We then profiled cell-cell communication within the AGM microenvironment [20]. We identified 142 unique signaling pathways, composed of 248 ligand-receptor interactions, that target HECs (*p* < 0.05, Appendix A). The calculated ‘probability’ quantifies the relative strength of each interaction compared to all other interactions, providing a way to interpret interaction strengths in the case of extreme *p* values.

Several of these signaling modalities were previously known to support EHT and HSC formation, including TGFβ, Notch, and Wnt (Figure 1C and Appendix A)**.** We specifically identified interactions between NOTCH1 + JAG1, NOTCH1 + DLL4, and TGFb + TGFBR1 that have been functionally shown to facilitate EHT [21,22,23] (Appendix A). In some cases, these interactions were more prominent in HECs compared to vascular endothelial cells (Figure 1C and Appendix A). However, some ligands produced and/or received at moderate levels by many AGM cell populations, making it difficult to ascertain targeted cell-cell communications. For example, TGFβ receptor expression and communication could potentially occur via ligands produced in several cell types (Figure 1D,E).

### 2.2. Interactions with Stromal Cells and Extracellular Matrix Impact EHT

We focused on cases in which HECs were receptive to ligands produced by specific vascular endothelial cells, pericytes, and/or other stromal populations in the AGM environment (Figure 1F–K). This approach allowed us to better validate communication modalities via functional evidence and/or targeted bioinformatics strategies. For example, CXCL12 produced by fibroblasts and pericytes is specifically received by CXCR4 on HECs (Figure 1F–H). CXCL12-CXCR4 interactions are required for HSC formation [10,24].

We also identified cell interactions between extracellular matrix (ECM) proteins and receptor proteins that were remarkably specific to HECs. For example, the ADGRG6/GP126 receptor which was only appreciably expressed in HECs (Figure 1I,J). ADGRG6 interactions with collagen ligands (COL4A1 or COL4A2) were predominant, as opposed to alternative ligands (PRNP or COL4A5, Figure 1J,K). ADGRG6 receptor interactions with collagen can induce expression of *GATA2* and *VEGFR2/KDR* [25] and GATA2 activities can promote EHT [26]. We noted relevant upstream and gene induction downstream of ADGRG6 signaling in HECs (Figure 1J).

Other ECM-Receptor interactions were less specific, although Laminin signaling modalities were enriched in HECs [27] (Appendix A). Laminin ligands were produced by several endothelial and stromal populations in the AGM, including HECs themselves. While expression of some Laminin receptors were enriched in HECs vs. vascular endothelial cells (e.g., SV2C), the strongest signaling was predicted to occur between Laminins and more widely expressed ITGA9/ITGB1 integrin receptors (Appendix A).

These findings confirm and extend the repository of signaling mechanisms that impact HEC development within the AGM niche and lend insight into specific signaling inputs from vascular endothelial and stromal cells that support developmental hematopoiesis. Indeed, some degree of retention in the AGM endothelial lining has been implicated in the ‘arterial education program’ necessary to produce engraftable hematopoietic stem cells [24]. However, endothelial retention is ultimately counterproductive to HSC release into circulation. This led us to consider mechanisms responsible for the release of nascent HSCs (or clusters thereof) from the aortic endothelial lining.

### 2.3. Inflammatory Signaling from Preformed Circulating Hematopoietic Cells Can Direct HSC Release

Sterile inflammatory signaling directs HEC specification, EHT, and HSC formation in zebrafish [2], mouse [11], and cultured human cells [13]. We found that TNF produced by macrophages and granulocytes could interact with receptors on human HECs, in alignment with prior findings in model systems (Figure 2A). While TNF signaling via TNFR1 was equivalent across all HECs and vascular endothelial cells, only HECs and (to a lesser extent) HSCs expressed TNFR2 (Figure 2B,C). TNFR2 is required for AGM hematopoiesis in the zebrafish [2] and can initiate different signaling than TNF-TNFR1 binding [28].

We looked for evidence of downstream activation of TNF-responsive transcription factors within the data. Surprisingly, acute inflammation and other inflammatory pathways were not statistically significantly enriched in HEC vs. vascular endothelium by gene set enrichment analysis (GSEA). For example, the Inflammatory Response pathway (GOBP) was not statistically enriched in HECs vs. vascular endothelium (NES 1.1, *p_ad_*_j_ = 0.81). We instead identified alternative transcriptional responses to TNFα, including STAT3 target activation in HEC (NES = 1.75 vs. vascular endothelium, *p_adj_* = 0.036, Figure 2D). In an orthogonal experiment, we used decoupleR to identify transcriptional regulatory networks that were active among sampled embryonic cells. This experiment also identified STAT3 signaling upregulation in HECs vs. vascular endothelial cells (Figure 2E). In addition to mediating TNFα-induced inflammatory cytokine production in some circumstances, STAT3 signaling facilitates vascular endothelial permeability and cell migration in the context of tumorigenesis [29,30]. Thus, an inferred effect of TNF-TNFR2 signaling in HECs may be to promote endothelial permeability and EHT/HSC escape.

We further identified regulatory interactions between hematopoietic cells and HECs that have been directly implicated in cell migration and escape from adherent endothelial environments. These signaling pathways included Osteopontin (SPP1/OPN) [31,32,33] (Figure 2F). Osteopontin signaling was mediated by macrophage-derived ligand, although several cell types produce SPP1 (Figure 2F,G). Several integrins can serve as SPP1 receptors, but ITGAV and ITGA5 were notably more highly expressed in HECs vs. vascular endothelium (Figure 2G). A key downstream response to Osteopontin signaling is the induction of matrix metalloproteases (MMPs) [31,32]. In HECs, MMPs facilitate HEC release from the arterial endothelium [34]. Indeed, the expression of several MMPs was enriched in HECs vs. vascular endothelium (Figure 2H). While MMP2/9 facilitate EHT in mice, it is likely that analogs more highly expressed in these HECs, such as MMP2, MMP16, and MMP28, mediate this process during human embryonic development.

These experiments begin to paint a systemic portrait of factors that together help initiate developmental hematopoiesis, including stromal support networks and interactions with circulating hematopoietic factors, including a role for inflammatory signaling in the breakdown of the endothelial barrier that would be necessary to release nascent HSCs into circulation.

### 2.4. Systemic Analysis Identifies Fetal Liver-Derived Support for HSC Formation and Release

We next extended the scope of potential interactions to include the fetal liver, the site of initial engraftment for HSCs produced in the AGM region [7]. We hypothesized that the fetal liver might secrete hematopoiesis-inducing factors to indicate its readiness to receive HSCs from the AGM, enabling developmental coordination to promote engraftment in a similar process to bone marrow during later development [35,36]. Indeed, our initial analyses indicated potential liver-derived F2/Thrombin signaling to ADGRG6 on HECs [37] (Figure 1F).

To create a multi-organ single cell data set, we integrated and annotated data from fetal liver and AGM cells, identifying hepatocytes, stellate cells, and activated stellate cell populations among fetal liver cells [38,39] (Figure 3A,B). We then used the full integrated object across all time points to identify 73 unique signaling pathways, composed of 232 ligand-receptor interactions, that targeted HECs (*p* < 0.05, Appendix A).

We noted strong Protein C (PROC) and protease-activated receptor (PAR) signaling among fetal liver-HEC interactions (Figure 3C–H). In addition to roles in coagulation, PROC and PAR signaling regulate EMT and the regulation of endothelial integrity [40,41]. PROC was exclusively produced by fetal liver hepatocytes, and HECs expressed increased levels of the PROC Receptor (PROCR/EPCR) compared with vascular endothelial populations (Figure 3C,D). PROC signaling induces EMT in breast cancer cells [40], and EPCR is an established HSC marker that promotes engraftment following irradiation [42,43,44]. HECs showed 8-fold higher EPCR expression compared with HSCs, and furthermore co-expressed F2R/PAR1 that is necessary for EPCR to mediate PROC-dependent effects [41] (Figure 3E,H). PAR signal ligands originating from fetal liver were also relatively specific for HECs and vascular endothelium (Figure 3F,G), with HECs expressing higher levels of F2R and PARD3 than vascular endothelial cells (Figure 3G,H). These signaling interactions may also be involved in breakdown of the AGM endothelial lining, as has been shown in other circumstances [45].

Finally, we identified novel signaling modalities that may link fetal liver development as a receptive organ to developmental hematopoiesis and EHT. Fetal liver-derived Angiopoietin-like 4 (ANGPTL4) was specifically received by HECs (Figure 3I). ANGPTL4 induces vascular leaks by destroying tight junctions in the endothelial lining [9]. ANGPTL4 binds to ITGA5B1, which is increased in HECs vs. vascular endothelial cells (Figure 3J). In turn, this activates PAK1/Rac, Notch, and Wnt signaling, as well as dissociation of Claudin 5 and VE-Cadherin (CLDN5, CDH5) at tight junctions [9]. ANGPTL4 can then bind free CLDN5 and CDH5, preventing tight junction restoration. The liver secretes cleaved ANGPTL4 (cANGPTL4), the most potent version of this gene product that has been linked to cancer metastasis [9]. In addition to other mechanisms, this represents a novel mechanism by which the fetal liver could plausibly enhance the release of nascent HSCs from the endothelial wall and into circulation (Figure 4).

## 3. Discussion

In this study, we identified known and putative signaling inputs that support human hematopoiesis via cell autonomous processes and autocrine signaling, as well as supportive interactions with local stroma and circulating inflammatory cells. Our findings extend insights into cell-intrinsic mechanisms that facilitate EHT, including the recently elucidated contributions of the SPI1-KLF1/LYL1 signaling axis and a cis-regulatory network of transposable elements [14,15]. We identified interactions within the AGM mirror the systemic factors and microenvironment that regulates hematopoiesis in the bone marrow [49], and provide an inclusive view of the factors supporting definitive hematopoiesis in the mammalian embryo. Our findings are consistent with the notion that production of HECs and/or HSCs is a systemic process involving cells outside of the AGM endothelium, including at minimum signals from the fetal liver that may coordinate HSC arrival. Similarly, the fetal bone marrow produces CXCL12 and KITLG to induce migration towards marrow [50].

Many of the signaling pathways required for EHT are directly involved with the EMT and associated with cancer progression [6], including mechanisms by which liver signals induce EMT and promote liver metastases [51]. These parallels are also reflected by gene set enrichment analysis that identified ‘tumor angiogenesis’, but not canonical angiogenesis, as enriched in HECs vs. vascular endothelium (Figure 2D). Many of the signaling modalities linked to EHT impact cell morphogenesis and HEC “escape” from the arterial wall, including the dissolution of cell junctions, eliminating apico-basal cell polarity, and adopting a mesenchymal (non-adherent) gene expression pattern. Our findings related to ANGPTL4 and PROC/PAR signaling may reflect discrete mechanisms by which local and systemic signaling might help promote EHT (Figure 4).

This study focused on samples collected in temporal correlation with EHT and primary HSC expansion in the AGM. The data sets used in our analysis did not include fetal liver HSCs, likely because ‘secondary’ HSC expansion in the fetal liver occurs later in gestation [46]. Secondary expansion greatly increases HSC quantities and endows HSCs with properties necessary for bone marrow engraftment [46,47,48]. While our results did identify some signaling processes known to permit fetal liver HSC expansion (e.g., IGF2 and ANGPTL3 signaling [48]), these should be interpreted with caution. Targeted interrogation to discover systemic signals and regulatory cell types that facilitate fetal liver HSC expansion will form the basis for future studies.

Our results further link inflammatory signaling to EHT. Osteopontin (OPN), encoded by the *SPP1* gene, impacts cell migration and cancer metastasis [31,32]. HECs had increased expression of the OPN-binding integrins AVB1 and A5B1 compared to vascular endothelial cells. ITGAVB1 causes human breast cancer cells to undergo chemotaxis towards OPN gradients, and induces glioma metastasis [52]. We also identified downstream targets of OPN signaling, including increased expression of matrix metalloproteases that mediate cleavage of the extracellular matrix during metastasis. MMP2/9 was highly expressed HECs compared to vascular endothelium. MMP2/9 was previously shown to regulate EHT in murine models [34]. It is likely that analogous MMPs participate in this process during human development (Figure 2H).

More broadly, our findings align with recent studies showing how hematopoietic cells participate in HEC and HSC formation through the production of sterile inflammation. TNF ligand, produced in neutrophils and macrophages and received by HEC, causes downstream NFκB signaling. This inflammatory signaling mechanism is important for murine [11], zebrafish [2,12] and in vitro HSC formation [13]. We paradoxically noted downregulation of the TNFR1-induced responses in HECs vs. vascular endothelium, likely because of the importance of TNFR2-induced response that is necessary for HEC specification and maintenance [2].

Our findings indicate that HECs produce basement membrane ligands, like Collagen and Laminins, which participate in autocrine and local signaling interactions. These results may provide novel insight into the biology underlying EHT, specifically the processes by which arterial endothelial cells fill in gaps left following EHT. Extracellular matrix proteins, including Laminins, may help neighboring vascular endothelial cells to migrate and fill in arterial wall space left by migrating HSCs. Arterial wall maintenance is necessary to avoid vascular disintegration and hemorrhage that occurs when HEC specification is enhanced via enforced Runx1^+^ expression [53]. The mechanobiology that supports AGM hematopoiesis, including mechanisms that maintain endothelial integrity during EHT and the release of nascent HSCs, comprises an exciting area for functional interrogation.

This study is based on single cell transcriptomics data and established ligand-receptor databases [20]. A similar approach has identified important cell communication modalities in physiologic and pathophysiologic microenvironments [54]. The identification of established control signaling pathways, and links to functionally validated mechanisms, gave us confidence in the presented findings. However, the scope of our experiments was limited to established ligand-receptor databases [20]. Future spatial transcriptomics and functional validation experiments will further interrogate the importance of specific cell-cell interactions in the developing embryo. Targeted studies may also focus on rare subpopulations that have been functionally linked to developmental hematopoiesis—e.g., a population of Ng2^+^Runx1^+^ perivascular smooth muscle cells recently shown to promote HEC and HSC formation [3].

Defining these active signaling pathways in the complex in vivo hematopoietic niche is important for understanding normal and diseased hematopoiesis, as well as reconstituting hematopoiesis in vitro to produce blood cell-based transfusion products, clinical testing reagents, and other off-the-shelf blood cell-based therapeutics. Indeed, recent studies leveraged primary human embryonic single cell data to develop engraftable HSCs from induced pluripotent stem cells [16,17]. Prior studies have aimed to recapitulate stromal and mesenchymal cells, which are typically absent from in vitro models. Signaling from pre-formed hematopoietic cells and/or fetal liver may augment in vitro inefficiencies and elucidate new developmental paradigms that influence developmental hematopoiesis.

## 4. Materials and Methods

### 4.1. Data Collection and Processing

Single Cell RNA data were obtained from Gene Expression Omnibus (GEO) under accession GSE162950 [4]. We combined AGM4.5, AGM5a, AGM5b, and AGM6 datasets using Harmony to create our completed AGM object which acted as a basis for all future analysis. The Fetal Liver + AGM object was created by using Seurat (v5) default tools for combining Seurat objects [18]. Seurat objects representing fetal liver cells at 4.5, 5, and 6 weeks were combined with vascular endothelial cells, HEC, HSC, and megakaryocyte AGM cells using the merge function. We restricted analysis to the same study and data sets from Calvanese et al. [4] to limit batch/technical effects as able.

Quality control metrics generally aligned with the original manuscript [4], including 200–8000 features with <10% mitochondrial DNA content. Data were normalized and processed using Seurat (v5.0.1) package functions (NormalizeData, FindVariableFeatures, ScaleData, RunPCA, FindNeighbors, FindClusters, and RunUMAP).

We determined cell and cluster annotations using SingleR (Human Primary Cell Atlas and Bluepoint Encode), CellTypist, ScType, manual curation based on key marker genes, and published annotations [4]. Cells considered to be placental cells were removed. All markers and parameters are found in the supplemental code section and at Github. Statistical comparisons for selected gene expression were obtained from the Seurat FindMarkers function, which represent statistical estimates after adjusting for multiple comparisons using the Bonferroni correction [18].

### 4.2. Cell Interaction Analysis

We used CellChat (v2.1.1) with standard parameters to identify interactions between cells [20]. We typically examined interactions between individual clusters, rather than cell types, to improve discovery power. Analyses of intercellular signaling was performed at the level of cell clusters (e.g., Endo1, Endo2, Endo3) or cell types (e.g., Endo) as indicated. In some cases, cluster level analysis was necessary due to heterogeneity among subgroups within each cell type. We mandated that >25% cells in a group participated in each ligand-receptor interaction. For simplicity, figures and raw output (Appendix A) are displayed at the cell type level unless otherwise. “TriMean” was used as the argument for the computeCommunProb function. We analyzed interactions with four possible annotations (“Secreted Signaling”, “Cell-Cell Contact”, “ECM-Receptor”, and “Non-Protein Signaling”). Statistical estimates for signaling pathways were obtained directly from CellChat output, which represent estimates made following permutation testing that randomly permutes cell group labels and recalculates communication probability (n = 100 permutations) [20]. Many interaction *p*-values were extremely low, so we also include calculations for the relative strength of each interaction compared to all other interactions (a value represented as “prob”) in Appendix A. CellChat functions were used to produce graphical output.

### 4.3. Gene Set Enrichment Analysis

The function “FindMarkers” from the package Seurat was used to generate a list of marker genes for each cluster of the AGM. Marker genes were tested against the “Hallmark”, “C2”, “C3”, and “C5” genesets from the Molecular Signature Database using the package fgsea. This calculated the Net Enrichment Score of each pathway based on the marker genes of the HE. We performed a similar analysis comparing HECs to ECs and HECs to HSCs with the “FindMarkers” function from Seurat.

### 4.4. Transcription Factor Activity

We assessed signaling pathway output via transcription factor activities using decoupleR. Seurat objects were converted into adata objects using ScCustomize and analyzed in a python3 environment (Ubuntu v22.02). Transcription factor activity inference was performed on the adata object using the CollectTRI database and analyzed using standard parameters.

### 4.5. Code Availability

All data are publicly available from Gene Expression Omnibus (GEO) under accession GSE162950. Analysis code and scripts are posted at https://github.com/biomain/AGM_EHT_scRNA (accessed on 23 December 2024), and available by request.

## 5. Conclusions

This study was designed to test whether in silico methods could detect known cell communications pathways that impact human developmental hematopoiesis, forming a basis for the discovery of novel factors that might also temporally impact HEC/HSC formation through local interactions in the AGM and/or distant signaling interactions initiated by other organs or tissues. Through the application of CellChat on a scRNA dataset during definitive hematopoiesis, we identified known and novels signals that impact developmental hematopoiesis. These findings elucidate a more complete portrait of the systemic and environmental signals that direct HEC and HSC formation during human embryonic development, including extracellular matrix interactions and direct contact with neighboring cells in the AGM region, as well as key interactions with circulating hematopoietic cells and long-range secreted signaling interactions with fetal liver.

## Figures and Tables

**Figure 1 ijms-26-00301-f001:**
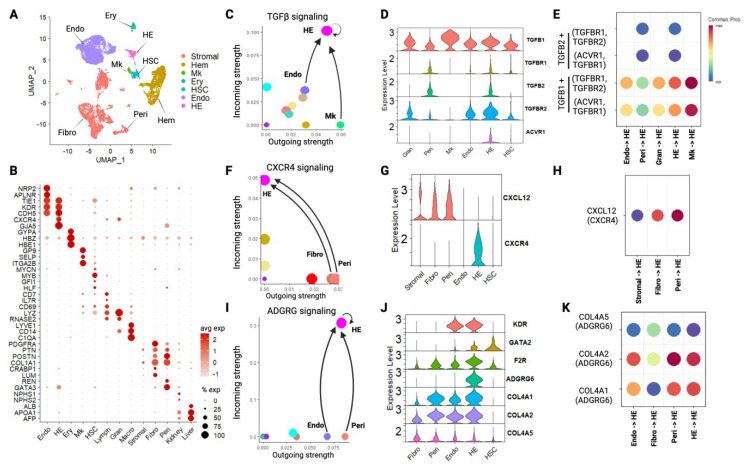
Identification of in vivo signaling mechanisms that regulate hemogenic endothelial (HE) development. (**A**) UMAP depicting populations of cells derived from human embryonic tissue. (**B**) Dot plot supporting annotation of human embryonic cell populations. (**C**) Scatterplot depicting outgoing and incoming Transforming Growth Factor β (TGFβ) signaling among analyzed human embryonic cells. HE cells participate in autoregulation and receive signals from ligands produced in vascular endothelial cells (Endo) and megakaryocytes (MKs). (**D**) TGFβ receptor abundance in granulocytes, pericytes, megakaryocytes, vascular endothelial cells, HE cells, and hematopoietic stem cell populations. (**E**) TGFβ ligand-receptor interactions received by HE cells, indicating strongest interactions between TGFβ1 ligand binding to receptor molecules composed of TGFβR1/TGFβR2 or TGFβR1/ACVR1 (Activin Receptor Type 1). *p* < 0.01 for all ligand-receptor interactions. (**F**) Scatterplot depicting CXC motif chemokine ligand 12 (CXCL12)-CXC motif chemokine receptor 4 (CXCR4) signaling activity in analyzed human embryonic cells. The strongest signaling interaction was between fibroblasts secreting CXCL12 and HE cells expressing CXCR4 receptor. (**G**) CXCR4 receptor expression in vascular endothelial and HE cell populations (HE vs. vascular Endo comparison: logFC = 3.3, *p_adj_* < 1 × 10^−76^, HE vs. HSC comparison: logFC = 3.0, *p_adj_* < 2 × 10^−76^). (**H**) CXCL12-CXCR4 ligand-receptor interactions received by HE cells indicate that the strongest signaling interactions occur from fibroblasts and pericytes to HE cells. *p* < 0.01 for all ligand-receptor interactions. (**I**) Scatterplot depicting Adhesion G Protein-Coupled Receptor G (ADGRG) signaling activity in analyzed human embryonic cells. HE cells participate in autoregulation and receive signals from ligands produced by pericytes and vascular endothelial cells. (**J**) Collagen ligand and ADGRG6 receptor expression in analyzed cell types. HE cells selectively express ADGRG6 receptor. HE cells, vascular endothelium, pericytes, and fibroblasts express relevant collagen ligands (ADGRG6 HE vs. vascular Endo comparison: logFC = 5.5, *p_adj_* < 1 × 10^−76^, HE vs. HSC comparison: logFC = 4.8, *p_adj_* < 3 × 10^−83^). (**K**) The strongest ADGRG6 communication probabilities support Collagen Type 4 alpha 1 (COL4A1)/Collagen Type 4 alpha 5 (COL4A5) ligand binding to ADGRG6 on HE cells. *p* < 0.01 for all ligand-receptor interactions. Cell type abbreviations: Ery, erythroid. Endo, endothelium (vascular). HE, hemogenic endothelium. Mk, megakaryocyte. HSC, hematopoietic stem cell. Fibro, fibroblast. Peri, pericyte. Hem, hematopoietic cell (mature). Mature hematopoietic cells include lymphoid (Lymph), granulocyte (Gran), and macrophage (Macro) populations.

**Figure 2 ijms-26-00301-f002:**
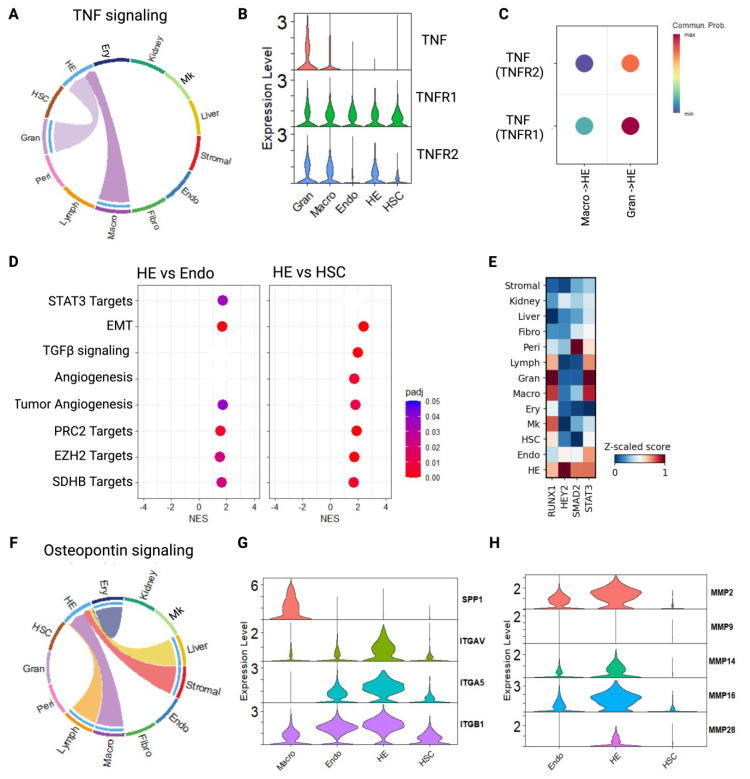
Circulating hematopoietic cells provide inflammatory signaling that enacts downstream transcriptional effects and coordinates nascent HSC escape from arterial endothelium. (**A**) Circle plot depicting Tumor Necrosis Factor (TNF) signaling input from granulocytes and macrophages received by HE cells. Colors (arbitrary) represent different cell types. (**B**) TNF Receptor 2 (TNFR2) expression is enriched in HE cells and hematopoietic cells, whereas TNFR1 expression is consistent across HE cells, vascular endothelium, HSCs, and mature hematopoietic cells (TNFR2 HE vs. vascular Endo comparison: logFC = 1.5, *p_adj_* < 9 × 10^−30^). (**C**) TNF-TNFR2 signaling is selectively active in HE cells. *p* < 0.01 for all ligand-receptor interactions. (**D**) Gene set enrichment analysis (GSEA) showing selected pathway induction in HE cells vs. vascular endothelium or HSCs. STAT3 activity is induced in HE cells vs. vascular endothelial cells and remains on in HSCs. NES, normalized enrichment score. (**E**) Heat plot depicting selected transcription factor-induced gene expression activities in analyzed human embryonic cells. STAT3 activity, a downstream result of TNF signaling, is induced in HE cells vs. vascular endothelium. Inflammatory NFκB is not. Other key transcription factor signaling activities that support developmental hematopoiesis (e.g., RUNX1, NOTCH/HEY2, SMAD2) are shown for comparison. (**F**) Osteopontin (SPP1) signaling activity in analyzed human embryonic cells. Ligand input from multiple cell types, including inflammatory cells, are selectively received by HE cell receptors. Colors (arbitrary) represent different cell types. (**G**) Expression of Osteopontin receptor molecules Integrin Alpha V (ITGAV) and Integrin subunit Alpha 5 (ITGA5) are enriched in HE cells vs. vascular endothelium, whereas all endothelial cells express Integrin subunit Beta 1 (ITGB1). ITGAV HE vs. vascular Endo: logFC = 1.5, *p_adj_* < 5 × 10^−38^, ITGA5 HE vs. vascular Endo: logFC = 1.2, *p_adj_* < 2 × 10^−55^. (**H**) Osteopontin signaling can induce expression of matrix metalloproteases (MMPs). MMP2 expression is enriched in HE cells vs. vascular endothelium and is known to facilitate escape of nascent HSCs from the arterial endothelial wall during developmental hematopoiesis. MMP2 HE vs. vascular Endo: logFC = 1.4, *p_adj_* < 2 × 10^−51^, MMP16 HE vs. vascular Endo: logFC = 1.8, *p_adj_* < 5 × 10^−78^, MMP28 HE vs. vascular Endo: logFC = 2.8, *p_adj_* < 6 × 10^−110^. Cell type abbreviations: Ery, erythroid. Endo, endothelium (vascular). HE, hemogenic endothelium. Mk, megakaryocyte. HSC, hematopoietic stem cell. Fibro, fibroblast. Peri, pericyte. Hem, hematopoietic cell (mature). Mature hematopoietic cells include lymphoid (Lymph), granulocyte (Gran), and macrophage (Macro) populations.

**Figure 3 ijms-26-00301-f003:**
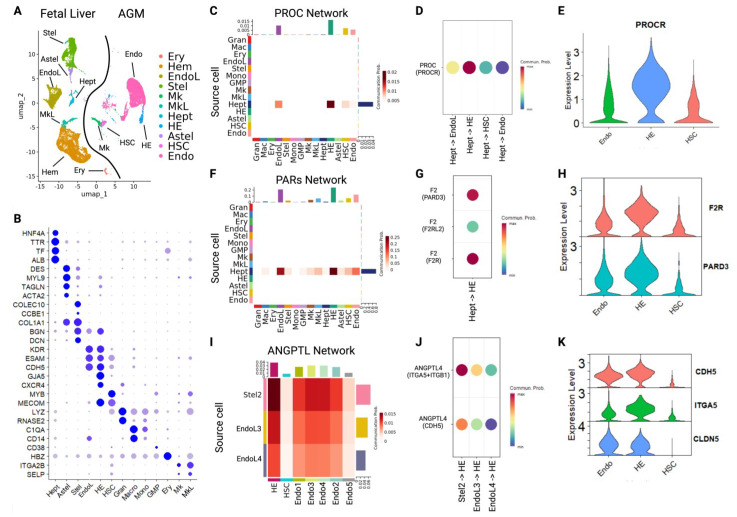
Long range interactions between fetal liver hepatocytes and AGM HE cells regulate developmental hematopoiesis. (**A**) UMAP depicting cell populations from human embryonic tissues, including vascular endothelium, HE cells, HSCs, and lineage-committed hematopoietic cells from the aorta-gonad-mesonephros region (AGM) along with hepatocytes, endothelium, stellate cells, activated stellate cells, and hematopoietic cell types from within the fetal liver sample. (**B**) Dot plot supporting cell type annotations from AGM and fetal liver populations. Larger dot reflects higher percent expression among the indicated cell type. Darker blue represents higher average expression. (**C**) Analysis of Protein C (PROC) signaling activities show selective secretion from fetal liver hepatocytes that is received by HE cells, vascular endothelial cells, and HSCs. (**D**) PROC-PROC Receptor (PROCR) interactions are strongest between fetal liver and HE cells, as compared to vascular endothelium and HSCs. *p* < 0.01 for all ligand-receptor interactions. (**E**) PROCR expression is significantly higher in HE cells as compared to vascular endothelium and HSCs. HE vs. vascular Endo: logFC = 1.7, *p_adj_* < 7 × 10^−78^. (**F**) Analysis of Protease-Activated Receptor (PAR) signaling activities shows that ligands secreted by fetal liver hepatocytes can be received by HE cells and vascular endothelial cells in the AGM. (**G**) Analysis of PAR signaling activities in HE cells shows predominant interactions between fetal liver-derived F2/thrombin ligand and thrombin receptor (F2R) or Par-3 Family Cell Polarity Regulator (PARD3) on HE cells. *p* < 0.01 for all ligand-receptor interactions. (**H**) Expression of F2R and PARD3 are enriched on HE cells compared to vascular endothelium and HSCs. F2R HE vs. vascular Endo: logFC = 1.2, *p_adj_* < 6 × 10^−69^. (**I**) Analysis of Angiopoietin-like (ANGPTL) signaling activities shows that ligands secreted by fetal liver endothelial cells and stellate cells can be received by HE cells in the AGM. (**J**) Analysis of ANGPTL signaling activities in HE cells shows predominant interactions between ANGPTL4 and Integrin (ITGA5/ITGB1) receptors on HE cells. *p* < 0.01 for all ligand-receptor interactions. (**K**) Expression of ligands and receptors for ANGPTL signaling in AGM cells. ANGPTL4 inhibits tight junctions by binding and sequestering Claudin 5 (CLDN5) and VE-Cadherin (CDH5). ITGA5 HE vs. vascular Endo: logFC = 1.2, *p_adj_* < 2 × 10^−55^. Cell type abbreviations: Ery, erythroid. Endo, endothelium (vascular). EndoL, endothelium from fetal liver sample. HE, hemogenic endothelium. Mk, megakaryocyte. MkL, megakaryocytes from fetal liver sample. HSC, hematopoietic stem cell. Hem, hematopoietic cell (mature). Stel, stellate cells. Astel, activated stellate cells. Hept, hepatocytes. Mature hematopoietic cells include lymphoid (Lymph), granulocyte (Gran), and macrophage (Mac), monocyte (Mono), and granulocyte-monocyte progenitor (GMP) populations.

**Figure 4 ijms-26-00301-f004:**
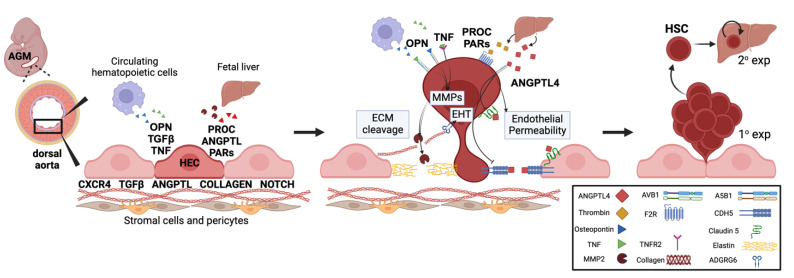
Summary diagram for selected signaling inputs to hemogenic endothelial cells (HEC), with resultant effects on development and ultimately nascent release from the aortic endothelial layer during developmental hematopoiesis. Shown are signals derived from stromal support cells (fibroblasts and pericytes), circulating hematopoietic cells (e.g., macrophages and granulocytes), and fetal liver-derived chemokines that impact the endothelial-to-hematopoietic transition (EHT), endothelial permeability, and extracellular matrix (ECM) cleavage, including matrix metalloproteases (MMPs). These developmental processes impact ‘primary’ (1°) hematopoietic stem cell (HSC) expansion in the aorta-gonad-mesonephros (AGM) region. These HSCs then migrate to the fetal liver, where they undergo secondary (2°) expansion and are prepared to colonize bone marrow. Key signaling processes that regulate secondary expansion have been reviewed elsewhere [46,47,48]. OPN, Osteopontin, TNF, Tumor Necrosis Factor. PROC, Protein C. ANGPTL, Angiopoietin-like. PARs, Protease-Activated Receptors. Created with BioRender.

## Data Availability

Analysis code and scripts are posted at https://github.com/biomain/AGM_EHT_scRNA and available by request.

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
