# Peer review of "Integrated Local and Systemic Communication Factors Regulate Nascent Hematopoietic Progenitor Escape During Developmental Hematopoiesis"

_ijms, 2024, doi:10.3390/ijms26010301_

Round 1

Reviewer 1 Report

Comments and Suggestions for Authors

The presented work was carried out in silico and represents a successful analysis of the available data - Single Cell RNA data were obtained from Gene Expression Omnibus (GEO) under accession GSE162950 [Calvanese, V.; Capellera-Garcia, S.; Ma, F.; Fares, I.; Liebscher, S.; Ng, E.S.; Ekstrand, S.; Aguadé-Gorgorió, J.; Vavilina, A.; 417 Lefaudeux, D.; et al. Mapping Human Haematopoietic Stem Cells from Haemogenic Endothelium to Birth. Nature 2022, 604, 418534–540,]. Authors combined AGM4.5, AGM5a AGM5b and AGM6 datasets using Harmony to create their completed AGM object which acted as a basis for all future analysis. Authors restricted analysis to the same study and data sets from Calvanese et al to limit batch/technical effects as able. They have discovered known and potential signaling inputs that contribute to hematopoiesis through self-regulatory processes and autocrine signaling, as well as through interactions with the local stroma and circulating inflammatory cells. These interactions within the AGM reflect the systemic factors and microenvironment that govern hematopoiesis in the bone marrow, providing a comprehensive perspective on the factors that support definitive hematopoiesis in mammalian embryos. The findings are consistent with the notion that production of HECs and/or HSCs is a systemic process involving cells outside of the AGM endothelium, including at minimum signals from the fetal liver that may coordinate HSC arrival. Similarly, the fetal bone marrow produces CXCL12 and KITLG to induce migration towards marrow.

In a broader context, the results are in line with recent research that demonstrates the role of hematopoietic cells in the development of HEC and HSC through the generation of sterile inflammation.

Interesting and useful work has been done, worthy of publication.

Note.

Page 1, lines 28 and 36 Link to review by Dzierzak, E.; Speck, N.A. Of Lineage and Legacy: The Development of Mammalian Hematopoietic Stem Cells. Nat Immunol 2008, 9, 129–136. It deals with work performed on mice. Further on, the authors describe data related to humans. This should be noted, otherwise “A third wave, which occurs in the dorsal aorta-gonad-mesonephros (AGM) region and other major arteries 4.5 weeks post conception, produces short term multipotent progenitor cells and long term hematopoietic stem cells” sounds very strange, since the time of complete gestation in mice is 18–21 days. And the following reviews (links 7 and 8) are already about Diverse model systems, including mouse, chick and zebrafish embryos.

Further on, the authors carefully indicate the species of the embryo.

In the introduction on page 2, lines 61-62, it is worth immediately indicating that the work will be about a human embryo.

Author Response

Reviewer 1

Comment 1: The presented work was carried out in silico and represents a successful analysis of the available data - Single Cell RNA data were obtained from Gene Expression Omnibus (GEO) under accession GSE162950 [Calvanese, V.; Capellera-Garcia, S.; Ma, F.; Fares, I.; Liebscher, S.; Ng, E.S.; Ekstrand, S.; Aguadé-Gorgorió, J.; Vavilina, A.; 417 Lefaudeux, D.; et al. Mapping Human Haematopoietic Stem Cells from Haemogenic Endothelium to Birth. Nature 2022, 604, 418534–540,]. Authors combined AGM4.5, AGM5a AGM5b and AGM6 datasets using Harmony to create their completed AGM object which acted as a basis for all future analysis. Authors restricted analysis to the same study and data sets from Calvanese et al to limit batch/technical effects as able. They have discovered known and potential signaling inputs that contribute to hematopoiesis through self-regulatory processes and autocrine signaling, as well as through interactions with the local stroma and circulating inflammatory cells. These interactions within the AGM reflect the systemic factors and microenvironment that govern hematopoiesis in the bone marrow, providing a comprehensive perspective on the factors that support definitive hematopoiesis in mammalian embryos. The findings are consistent with the notion that production of HECs and/or HSCs is a systemic process involving cells outside of the AGM endothelium, including at minimum signals from the fetal liver that may coordinate HSC arrival. Similarly, the fetal bone marrow produces CXCL12 and KITLG to induce migration towards marrow.

In a broader context, the results are in line with recent research that demonstrates the role of hematopoietic cells in the development of HEC and HSC through the generation of sterile inflammation.

Interesting and useful work has been done, worthy of publication.

Response 1: We appreciate your careful reading and positive impression of our study.

Comment 2: Page 1, lines 28 and 36 Link to review by Dzierzak, E.; Speck, N.A. Of Lineage and Legacy: The Development of Mammalian Hematopoietic Stem Cells. Nat Immunol 2008, 9, 129–136. It deals with work performed on mice. Further on, the authors describe data related to humans. This should be noted, otherwise “A third wave, which occurs in the dorsal aorta-gonad-mesonephros (AGM) region and other major arteries 4.5 weeks post conception, produces short term multipotent progenitor cells and long term hematopoietic stem cells” sounds very strange, since the time of complete gestation in mice is 18–21 days. And the following reviews (links 7 and 8) are already about Diverse model systems, including mouse, chick and zebrafish embryos.

Further on, the authors carefully indicate the species of the embryo.

Response 2: Thank you for raising this important point. In the revised manuscript, we have clarified that multiple model systems have been used to generate insights about these developmental processes – and cited relevant review manuscripts that incorporate observations from multiple organisms (Introduction, 2nd paragraph). We have also specified that the temporal estimates of events during gestation (e.g., “A third wave…”) relate to human embryonic development in the revised manuscript.

Comment 3: In the introduction on page 2, lines 61-62, it is worth immediately indicating that the work will be about a human embryo.

Response 3: We appreciate this suggestion as well! We added clarification to indicate that our study focused on human embryonic development. These changes will undoubtedly make it easier for readers to understand and interpret the paper.

Reviewer 2 Report

Comments and Suggestions for Authors

Mammalian blood cells originate from specialized 'hemogenic' endothelial cells in major arteries. During the endothelial-to-hematopoietic transition (EHT), nascent hematopoietic stem cells (HSCs) enter circulation, colonizing the fetal liver before migrating to the bone marrow. The mechanisms facilitating EHT and HSC release are incompletely understood but may involve signaling from neighboring cells, stromal support cells, and systemic factors. Certain matters need to be addressed before the decision on publication can be made.

 1.     The research indicates that the process of blood cell formation in the mammalian embryo is influenced by various systemic factors and microenvironments, which encompass the generation of hematopoietic endothelial cells and hematopoietic stem cells. The generation of HECs encompasses cells beyond the AGM endothelium, notably including the fetal liver. The research establishes a connection between inflammatory signaling and EHT, highlighting the role of osteopontin in influencing cell migration and the progression of cancer metastasis. This manuscript is well-crafted and offers a wealth of valuable information.

2.     AGM-derived human stem cells (HSCs) demonstrate an enhanced capacity for producing hematopoietic cells and exhibit long-term multilineage engraftment potential. This makes them highly significant for therapeutic applications in regenerative medicine, especially in the treatment of blood disorders via genome editing methods. The authors are encouraged to include images that illustrate the mechanisms of primary (AGM) and secondary (fetal liver) expansion of HSCs. Additionally, a more detailed comparison of the properties of HSCs from these two embryonic sources should be clarified in Figure 4.

Author Response

Reviewer 2

Mammalian blood cells originate from specialized 'hemogenic' endothelial cells in major arteries. During the endothelial-to-hematopoietic transition (EHT), nascent hematopoietic stem cells (HSCs) enter circulation, colonizing the fetal liver before migrating to the bone marrow. The mechanisms facilitating EHT and HSC release are incompletely understood but may involve signaling from neighboring cells, stromal support cells, and systemic factors. Certain matters need to be addressed before the decision on publication can be made.

Comment 1.     The research indicates that the process of blood cell formation in the mammalian embryo is influenced by various systemic factors and microenvironments, which encompass the generation of hematopoietic endothelial cells and hematopoietic stem cells. The generation of HECs encompasses cells beyond the AGM endothelium, notably including the fetal liver. The research establishes a connection between inflammatory signaling and EHT, highlighting the role of osteopontin in influencing cell migration and the progression of cancer metastasis. This manuscript is well-crafted and offers a wealth of valuable information.

Response 1: We appreciate your positive comments. We agree that this manuscript can provide new insights on the cell types and processes that direct developmental hematopoiesis.

Comment 2.     AGM-derived human stem cells (HSCs) demonstrate an enhanced capacity for producing hematopoietic cells and exhibit long-term multilineage engraftment potential. This makes them highly significant for therapeutic applications in regenerative medicine, especially in the treatment of blood disorders via genome editing methods. The authors are encouraged to include images that illustrate the mechanisms of primary (AGM) and secondary (fetal liver) expansion of HSCs. Additionally, a more detailed comparison of the properties of HSCs from these two embryonic sources should be clarified in Figure 4.

Response 2: Thank you for this suggestion. In the revised Discussion, we highlight the developmental importance of primary vs secondary HSC expansion. We have also indicated these distinct processes in our revised Figure 4. We note that secondary HSC expansion mechanisms are not necessarily captured in the single cell data analyzed for this manuscript, given the developmental timeline in which cells were isolated and analyzed. We analyzed fetal liver samples obtained at <6 weeks gestation, since our intention to focus on temporal correlations with primary HSC expansion from the AGM. We in fact did not identify any HSCs within the analyzed fetal liver data, likely since secondary expansion occurs later in gestation. This precluded analysis of signaling mechanisms underlying HSC expansion in the fetal liver in this manuscript, although this comprises an important avenue for future study. The additions to this manuscript will certainly help to contextualize these developmental paradigms for readers.

Reviewer 3 Report

Comments and Suggestions for Authors

In this study, the authors used single cell RNA sequencing analysis from human embryonic cells to identify relevant signaling pathways that support nascent hematopoietic stem cells (HSCs) release. The authors concluded that through the application of CellChat on a scRNA dataset during definitive hematopoiesis, they identified known and novels signals that impact developmental hematopoiesis.

Comments:

The authors has some concerns as follows:

1.     This is an interesting study. This study has opened as a preprint in bioRxiv. Nevertheless, recently, there are several similar studies published, such as Feng et al., 2024, Systematic single-cell analysis reveals dynamic control of transposable element activity orchestrating the endothelial-to-hematopoietic transition (https://doi.org/10.1186/s12915-024-01939-5), Qu et al., 2024, SPI1-KLF1/LYL1 axis regulates lineage commitment during endothelial-to-hematopoietic transition from human pluripotent stem cells (doi: 10.1016/j.isci.2024.110409), Fowler et al., 2024, Lineage-tracing hematopoietic stem cell origins in vivo to efficiently make human HLF+ HOXA+ hematopoietic progenitors from pluripotent stem cells (doi: 10.1016/j.devcel.2024.03.003.), and Ng et al., 2024, Long-term engrafting multilineage hematopoietic cells differentiated from human induced pluripotent stem cells (https://doi.org/10.1038/s41587-024-02360-7). In these studies, some were used the same Single Cell RNA data (obtained from Gene Expression Omnibus (GEO) under accession GSE162950) with the present study. Therefore, considering the similarity and comparison, the authors need to strengthen their Introduction and Discussion.

2.     The quantification and statistical analysis are lacking in this study. How to verify and statistically analyze the data should be clearly described.

3.     In all Figures, the full name for abbreviations can be clearly shown in the legends.

4.     Overall, this manuscript needs a revision before it can be accepted.

Author Response

Reviewer 3

In this study, the authors used single cell RNA sequencing analysis from human embryonic cells to identify relevant signaling pathways that support nascent hematopoietic stem cells (HSCs) release. The authors concluded that through the application of CellChat on a scRNA dataset during definitive hematopoiesis, they identified known and novels signals that impact developmental hematopoiesis.

Comments:

The authors has some concerns as follows:

Comment 1.     This is an interesting study. This study has opened as a preprint in bioRxiv. Nevertheless, recently, there are several similar studies published, such as Feng et al., 2024, Systematic single-cell analysis reveals dynamic control of transposable element activity orchestrating the endothelial-to-hematopoietic transition (https://doi.org/10.1186/s12915-024-01939-5), Qu et al., 2024, SPI1-KLF1/LYL1 axis regulates lineage commitment during endothelial-to-hematopoietic transition from human pluripotent stem cells (doi: 10.1016/j.isci.2024.110409), Fowler et al., 2024, Lineage-tracing hematopoietic stem cell origins in vivo to efficiently make human HLF+ HOXA+ hematopoietic progenitors from pluripotent stem cells (doi: 10.1016/j.devcel.2024.03.003.), and Ng et al., 2024, Long-term engrafting multilineage hematopoietic cells differentiated from human induced pluripotent stem cells (https://doi.org/10.1038/s41587-024-02360-7). In these studies, some were used the same Single Cell RNA data (obtained from Gene Expression Omnibus (GEO) under accession GSE162950) with the present study. Therefore, considering the similarity and comparison, the authors need to strengthen their Introduction and Discussion.

Response 1: Thanks for your positive comment. We have also noted interest from the scientific community in the developmental hematopoietic process, as evidenced by the recent manuscripts you mention. The data set common to all studies is indeed a wealth of information. Two of the recent studies (Fowler et al and Ng et al) used the primary single cell data from HSCs as a ‘gold standard’ by which to create iPSC-derived progenitor cells. Feng et al and Que et al have revealed novel signaling mechanisms during the endothelial-to-hematopoietic transition. Importantly, these latter studies interrogate cell-intrinsic pathways and processes that facilitate the developmental transitions that ultimately produce HSCs. In contrast, we endeavored to take a more holistic approach to identify factors and other cell types that participate in this important developmental process. The incorporation of other niche cell types into our developmental model adds orthogonal insights to these others recent studies. We have added these important citations and comparisons to existing literature in our revised Introduction and Discussion sections.

Comment 2.     The quantification and statistical analysis are lacking in this study. How to verify and statistically analyze the data should be clearly described.

Response 2: Thanks for this important suggestion. We have added statistical estimations for relevant comparisons in our revised manuscript, noting in the Methods sections how these statistical comparisons were made.

Comment 3.     In all Figures, the full name for abbreviations can be clearly shown in the legends.

Response 3: Thanks for this important point. We have defined abbreviations in all Figure legends, which will undoubtedly improve the readability of our revised manuscript.

Comment 4.     Overall, this manuscript needs a revision before it can be accepted.

Response 4: We appreciate your suggestions and hope that our revised manuscript will be suitable for acceptance.

Round 2

Reviewer 3 Report

Comments and Suggestions for Authors

This revised manuscript has a great improvement and the reviewer has no further comments.